# Factors associated with memory of informal caregivers: A scoping review protocol

**Dingyue Wang**[1]*, **Sharron Rushton**[1], **Leila Ledbetter**[2], **Margaret Graton**[2], **Kimberly Ramos**[3], **Cristina C. Hendrix**[1,4]

1 School of Nursing, Duke University, Durham, NC, United States of America, 2 Duke University Medical Center & Archives, School of Medicine, Durham, NC, United States of America, 3 Des Moines University, West Des Moines, IA, United States of America, 4 GRECC Durham Veterans Affairs Medical Center, Durham, NC, United States of America

* dingyue.wang@duke.edu

**Data Availability Statement:** No datasets were generated or analyzed during the current study. All relevant data from this study will be made available upon study completion.

## Abstract

The ability to retain and remember information (memory) is essential to caregiving tasks. There is evidence that caregivers are at greater risk for experiencing deteriorations in cognitive status than non-caregivers, especially memory; however, we have a limited understanding of factors that are related to changes in caregiver memory. This scoping review intends to comprehensively map factors related to caregiver memory reported in the literature within the chronic caregiving context. Specific aims include (1) identifying factors related to caregiver memory; (2) examining how caregiver memory has been measured; and (3) describing changes in caregiver memory during their caregiving period. This review will be conducted following Arksey and O'Malley's framework and reported using the PRISMA guidelines for Scoping Reviews (PRISMA-ScR). Studies will be included if (1) the studies focus on home-based unpaid long term family caregiving; (2) study participants (patients), of any age, have one (or more) chronic illness or disability and receive care from a caregiver for 6 months or more; (3) caregivers are adults (> = 18 years of age). Any chronic disease or condition will be included. The search will encompass gray literature and peer-reviewed literature in MEDLINE (via Ovid), CINAHL Plus with Full Text (via EBSCOhost), Embase (via Elsevier), APA PsycINFO (via EBSCOhost), Sociology Source Ultimate (via EBSCOhost), and ProQuest Dissertations and Theses Global. Data extraction will include specific details about the participants, concept, context, study methods, and key caregiver-related findings. The Caregiver Health Model will provide a framework to categorize factors that impact caregivers' memory including caregiver health promotion activities, caregiver attitudes and beliefs, caregiver task, and caregiver needs. Factors that do not fall into the Caregiver Health Model domains will be organized by emerging themes.

## Introduction

Informal caregivers are relatives, partners, friends, or other individuals who provide uncompensated care to another person [1]. Caregiving is an important public health issue, and the

**Funding:** The author(s) received no specific funding for this work.

**Competing interests:** The authors have declared that no competing interests exist.

escalating need for care is a global concern due to various factors, such as a rising population living into advanced ages and a consequent surge in individuals with chronic conditions [2]. Approximately 18.2% of the US adult population, or 43.5 million informal caregivers, deliver care to an adult or child with special needs each year [3, 4]. Caregivers assist persons who have chronic illnesses or disabilities with a variety of care activities, as well as often playing a fundamental role in symptom management [5]. Nearly all caregivers help manage household tasks, such as shopping, finances, preparing meals, transportation, and home maintenance [6]. More than 50% of caregivers help with personal care and mobility tasks, such as managing medications, walking, feeding, dressing, or bathing [6]. Caregivers require the ability to understand the care that is needed and execute the skills that are required for caregiving tasks. This means that in addition to their physical capacity, caregivers need intact cognitive skills, such as memory, to complete their caregiving responsibilities safely and successfully [7].

Evidence suggests that caregivers face a greater risk for experiencing deteriorations in their cognitive status than non-caregivers [7–9]. Caregiving can be demanding and stressful. Many caregivers are often unprepared to manage the physical and emotional demands associated with caregiving, which may lead to cognitive problems [9–12]. Approximately 12.6% (11.7%-13.5%) of caregivers reported subjective cognitive problems, such as memory loss and more frequent confusion, compared to 10.2% of non-caregivers (p<0.001) [9]. When the care-receiver has dementia, spousal caregivers were reported to have higher risks of cognitive decline in global cognition, executive function, and language [7]. Additionally, compared to caregivers whose spouses were dementia-free, spousal dementia caregivers had a six-times higher chance of developing dementia, which can be partially explained by the substantial chronic stress experienced by caregivers [8]. Not only does cognitive decline impairs the performance of caregivers in their caregiving, but it also compromises caregivers' quality of life and functional ability in daily activities [9]. Although a number of interventions designed to decrease depression, improve health habits, and sustain physiological health in caregivers have been tested, no evidence-based approaches have been directly aimed at preventing or reducing cognitive problems in caregivers [12]. Memory is the most complex and multifaceted cognitive domain that requires specific attention [13]. Relative to non-caregivers, caregivers performed worse in memory tasks, including free recall, verbal memory, and Digit Span Backwards, which assesses the ability to retain and manipulate information in working memory [8, 14, 15]. Even after caregiving activity had ceased, further memory deterioration occurred [14].

A preliminary search of MEDLINE (via PubMed) and the Cochrane Database of Systematic Reviews was conducted on this topic. The reviews related to cognition, or memory specifically, all focused on non-caregiver populations [16–18]. No current or registered reviews on the topic of caregiver memory were identified. Thus, this scoping review can provide valuable information for theory development in caregiver cognitive health. We intend to determine the factors within the context of caregiving that are related to caregiver memory changes and how caregiver memory has been measured in the literature. As we aim to determine the range of a body of literature on caregiver memory clearly indicating the volume of literature available, a scoping review is more suitable than a systematic review [19].

The Caregiver Health Model (CGHM) will provide the conceptual framework for this scoping review [20]. The CGHM places family caregivers within the context of the external environment and identifies five constructs: caregiver health as a dependent construct and four independent constructs referred to as determinants, such as caregiver health promotion activities, caregiver attitudes and beliefs, caregiver tasks, and caregiver needs. This scoping review mainly focuses on caregiver memory for caregiver health as the dependent construct. In the present study, "caregiver health promotion activity" is defined as the strategies caregivers take to maintain or improve their memory (e.g., physical exercise, diet) and interventions or

programs designed for caregivers that are related to caregiver memory (e.g., caregiver training program). "Caregiver attitude and belief" is defined as caregivers' subjective evaluations and cognitive stances towards their caregiving responsibilities, the care recipients (patients), and the broader healthcare environment (e.g., cognitive stress appraisal). "Caregiver tasks" is defined as the activities (e.g., assisting with Activities of Daily Living (ADL)) or workload (e.g., hours of caregiving per day) that caregivers have that are related to caregiver memory. Finally, "caregiver needs" is defined as the specific requirements, challenges, and support mechanisms required by caregivers that is related to caregiver memory (e.g., emotional support needs, respite care needs). The factors related to caregiver memory revealed through this scoping review will be organized based on the four independent determinants mentioned above. Factors that do not fall into CGHM domains will be organized by emerging themes.

Successful completion of this review will contribute essential knowledge to direct further research in the development of interventions aimed at preventing memory decline and promoting healthy cognition in family caregivers. Identifying the factors related to caregiver memory can help us target interventions to halt or reverse progressively worsening memory decline in caregivers. Moreover, caregiver memory problems likely affect the quality and safety of care that caregivers can provide. Understanding caregivers' memory is critical to maintaining the independence and well-being of the caregiving dyads.

## Materials and methods

The proposed scoping review will be conducted in accordance with the framework from Arksey and O'Malley: identifying the research question, identifying relevant studies, study selection, charting the data, collating, summarizing, reporting results, and conducting consultation [21]. The results will be reported according to the Preferred Reporting Items for Systematic Reviews and Meta-Analyses extension for scoping reviews (PRISMA-ScR) (see S1 and S2 Checklists) [22, 26]. This scoping review protocol is registered to OSF (Registration DOI: https://doi.org/10.17605/OSF.IO/2V5XC).

### Stage 1: Identifying the research question

The objective of this review is to identify the available evidence about memory changes in informal caregivers. We intend to (1) identify the patient and caregiver factors related to caregiver memory; (2) examine how caregiver memory is measured; and (3) describe changes in caregiver memory during their caregiving period available in the literature. We are interested in identifying factors in the literature that have been shown to impact caregiver memory (e.g., caregiver stress), or can be influenced by caregiver memory (e.g., caregivers' quality of life).

### Stage 2: Identifying relevant studies

The Population Concept Context (PCC) framework is used to guide the development of the inclusion and exclusion criteria [23], which are presented in Table 1. Six electronic databases, MEDLINE (via Ovid), CINAHL Plus with Full Text (via EBSCOhost), Embase (via Elsevier), APA PsycINFO (via Ebscohost), Sociology Source Ultimate (via Ebscohost), and ProQuest Dissertations and Theses Global (via Ebscohost) will be reviewed. All databases will be searched from date of inception. The initial search was developed and conducted by a medical librarian (LL), with input from the other authors, and included a mix of keywords and subject headings representing informal caregiving and memory. The initial searches were conducted on 23 May 2022 with an updated search on 12 Jan 2023, and most recent search on 12 June 2023. We found 7,578 citations after removing duplicates. The full, reproducible search strategies for all included databases are in Table 2. The reference list of all articles selected for

**Table 1. Inclusion and exclusion criteria.**

| Participants | Concept | Context(s) |
|---|---|---|
| A caregiver is based on self-report, not by genetic or familial relationship. There will be no restriction on caregivers' relationship to the patient, gender, race, socioeconomic, or educational status. Caregiving tasks include, but not limited to, ADLs care, Instrumental Activity of Daily Living (IADL) assistance, and emotional support. Studies will be included if (1) study participants (patients), of any age, have one (or more) chronic illness(es) or disability(es) and receive care from a caregiver for 6 months or more; (2) caregivers are adults (>18 years of age). Studies will be excluded if (1) studies have no description of patient characteristics; (2) caregivers are professional caregivers; (3) caregivers are younger than 18 years of age; and (4) caregivers are not taking care of patients with chronic illness or disability (e.g., healthy child parental caregiving). | Guided by the CGHIM conceptual framework, the concepts of interest are caregiver memory and factors that may influence caregivers' memory, which may fall under caregiver health promotion activities, caregiver attitudes and beliefs, caregiver tasks, and caregiver needs. Memory is defined as the process of maintaining information over time [13]. Studies will be included if the study (1) provides information pertaining to factors related to memory of informal caregivers; (2) includes caregiver memory measurement tool (s); and (3) describes changes in caregiver memory during their caregiving period. Studies will be excluded if (1) caregiver memory is not measured and (2) the study solely focuses on other cognitive variables, outside of memory. | Context will be home-based unpaid long term (6 months or more) family caregiving, except healthy child parental caregiving. Caregiver (s) do not necessarily have to reside with the care recipient (s). Context for inclusion in this review will not be restricted by country, geographical location, language, or date to enable the full extent of available evidence to be synthesized. |

inclusion will be screened for additional relevant articles. The purpose of this scoping review is to obtain a broad range of relevant literature, including gray literature, in the field related to caregiver memory. Therefore, the date range will not be limited, with all aspects of social determinant of health factors included (e.g., race and language). This scoping review will include experimental as well as quasi-experimental study designs, including before-and-after studies, controlled trials (randomized and non-randomized), and interrupted time-series studies. Additionally, analytical and descriptive observational studies will meet inclusion, including case series; case-control studies; cohort studies (prospective and retrospective); cross-sectional studies (analytical and descriptive); and individual case reports. Reviews with a systematic methodology or meta-analysis will also be considered. Reference and citation tracking will be done using Citation Chaser on all articles included for data extraction [24].

## Stage 3: Study selection

After the search, all identified studies will be uploaded into Covidence (Covidence systematic review software, Veritas Health Innovation, Melbourne, Australia. Available at www. covidence.org), a software system for managing systematic reviews. Screeners will pilot 25 titles and abstracts to assess the clarity of eligibility criteria, the consistency of the criteria interpretation by screeners on the review team, and the need for refining the inclusion and exclusion criteria. Eligible full text will also be independently screened in detail against the inclusion criteria by two independent reviewers after a full text pilot. Full-text articles or documents that do not meet the criteria for inclusion will be excluded, and reasons will be provided in the final review report in accordance with journal guideline. The results of the search will be reported in full in the final report and presented in a PRISMA flow diagram in line with international standards (Fig 1) [25]. Conflicts will be resolved through discussion or a third reviewer.

## Stage 4: Charting the data

Data will be extracted from sources included in the scoping review by one reviewer and verified by another reviewer. A standardized form will be used to extract data. The data extracted will include specific details about the participants, concept, context, study methods, and key

**Table 2. Keywords and queries for search strategy of caregiver cognition.**

| Database / Study Registry (including vendor/platform) | Search | Query PMID |
|---|---|---|
| Medline (via Ovid), covers 1946-present | 1 caregivers | exp Caregivers/ |
| | 2 | ((Family OR informal OR unpaid OR friend) AND (carer* OR caregiver* OR caretaker*)).ti,ab. |
| | 3 | ((Family OR Informal OR unpaid OR friend) AND (Care Adj2 (giver* or giving or taker* or partner*))).ti,ab. |
| | 4 | 1 or 2 or 3 |
| | 5 memory | exp Memory/ or exp Mental Recall/ or exp Retention, Psychology/ or exp Repetition Priming/ |
| | 6 | (memory or memories or forget* or forgot* or recall* or remember* or recollect* or "mild cognitive impairment*" or priming).ti,ab. |
| | 7 | 5 or 6 |
| | 8 | 4 adj6 7 |
| CINAHL Complete (EBSCOhost), covers 1937-present | 1 caregivers | (MH "Caregivers") |
| | 2 | TI((Family OR Informal OR Unpaid OR friend) N5 (carer* OR caregiver* OR caretaker*)) OR AB((Family OR Informal OR Unpaid OR Friend) N5 (carer* OR caregiver* OR caretaker*)) |
| | 3 | TI((Family OR Informal OR Unpaid OR friend) N5 (Care W1 (giver* OR taker* OR partner*))) OR AB((Family OR Informal OR Unpaid OR friend) N5 (Care W1 (giver* OR taker* OR partner*))) |
| | 4 | S1 OR S2 OR S3 |
| | 5 memory | (MH "Memory"+) OR (MH "Memory, Short Term"+) OR TI (memory or memories or forget* or forgot* or recall* or remember* or recollect* or "mild cognitive impairment*" or priming) OR AB (memory or memories or forget* or forgot* or recall* or remember* or recollect* or "mild cognitive impairment*" or priming) |
| | 5 | S4 AND S5 |
| Embase (via Elsevier), covers 1947-present | 1 caregivers | Caregivers/de |
| | 2 | ((Family OR Informal OR Unpaid OR friend) NEAR/5 (Carer$ OR caregiver$ OR caretaker$)):ti,ab |
| | 3 | Family:ti,ab OR Informal:ti,ab OR Unpaid:ti,ab OR friend:ti,ab |
| | 4 | (care NEXT/2 (giver$ OR taker$ OR partner$)):ti,ab |
| | 5 | #3 AND #4 |
| | 6 | #1 OR #2 OR #5 |
| | 7 memory | 'memory'/de OR 'recall'/de OR 'repetition priming'/de OR 'forgetting'/de OR memory:ti,ab OR memories:ti,ab OR forget*:ti,ab OR forgot*:ti,ab OR recall*:ti,ab OR remember*:ti,ab OR recollect*:ti,ab OR "mild cognitive impairment*":ti,ab OR priming:ti,ab |
| | 8 | #6 AND #7 |
| APA PsycINFO (via Ebscohost), Journal coverage from 1800s – present, Citations & Summaries from 1600s - present | 1 caregivers | DE "Caregivers" OR DE "Caregiving" |
| | 2 | TI((Family OR Informal OR Unpaid OR friend) N5 (carer* OR caregiver* OR caretaker*)) OR AB((Family OR Informal OR Unpaid OR friend) N5 (carer* OR caregiver* OR caretaker*)) |
| | 3 | TI((Family OR informal OR unpaid OR friend) N5 (Care W1 (giver* or taker* or partner*))) OR AB((Family OR informal OR unpaid OR friend) N5 (Care W1 (giver* or taker* or partner*))) |
| | 4 | S1 OR S2 OR S3 |
| | 5 | DE "Memory" OR DE "Forgetting" OR DE "Priming" OR TI (memory OR memories OR forget* OR forgot* OR recall* OR remember* OR recollect* OR "mild cognitive impairment*" OR priming) OR AB (memory OR memories OR forget* OR forgot* OR recall* OR remember* OR recollect* OR "mild cognitive impairment*" OR priming) |
| | 5 | S4 AND S5 |

*(Continued)*

**Table 2.** (Continued)

| Database / Study Registry (including vendor/platform) | Search | Query PMID |
|---|---|---|
| Sociology Source Ultimate (EBSCOhost), covers 1908-present [WAITING ON VERIFICATION FROM EBSCO] | 1 caregivers | DE "CAREGIVERS" |
| | 2 | TI((Family OR Informal OR Unpaid OR friend) N5 (carer* OR caregiver* OR caretaker*)) OR AB((Family OR Informal OR Unpaid OR friend) N5 (carer* OR caregiver* OR caretaker*)) |
| | 3 | TI((Family OR informal OR unpaid OR friend) N5 (Care W1 (giver* or taker* or partner*))) OR AB((Family OR informal OR unpaid OR friend) N5 (Care W1 (giver* or taker* or partner*))) |
| | 4 | S1 OR S2 OR S3 |
| | 5 memory | DE "MEMORY" OR DE "PRIMING" OR TI (memory OR memories OR forget* OR forgot* OR recall* OR remember* OR recollect* OR "mild cognitive impairment*" OR priming) OR AB (memory OR memories OR forget* OR forgot* OR recall* OR remember* OR recollect* OR "mild cognitive impairment*" OR priming) |
| | 6 | S4 AND S5 |
| ProQuest Dissertations and Theses Global (via Ebscohost), covers 1861- present | 1 caregivers | NOFT((Family OR informal OR unpaid OR friend) N/5 (carer? OR caregiver? OR caretaker?)) |
| | 2 | NOFT((Family OR informal OR unpaid OR friend) N/5 (care PRE/1 (giver? OR giving OR taker? OR partner?))) |
| | 3 | S1 OR S2 |
| | 4 | NOFT(memory OR memories OR forget* OR forgot* OR recall* OR remember* OR recollect* OR "mild cognitive impairment*" OR priming) |
| | 5 | S3 AND S4 |

Note: S1 OR S2 means that the search results will include studies that match either S1 or S2 or both. It broadens the search to capture a larger set of relevant studies. S1 AND S2 means that the search results will include studies that satisfy both S1 and S2. This narrows down the search to find studies that meet both criteria.

findings relevant to the review questions. This data extraction form will be initially tested by two independent reviewers on five articles to check that all relevant information relating to the review questions is extracted. Each article will be organized to include the study characteristics (author, year, country, study design, study setting, study purpose, theoretical framework, and number of participants); characteristics of the caregivers and care recipients including number and demographics (e.g., age, race and ethnicity, relationship of caregiver-recipient dyads, caregiving tasks/intensity/duration of caregiving activity, etc.); memory (e.g., caregiver memory type, caregiver memory measurement tools/instruments, psychometrics of memory measurement tools/instruments, and the result of measurement); identified factors associated with caregiver memory (factor name, measurement tool/instrument, analytical approach, and factor result); and the main memory-related results of the study. The draft data extraction tool will be modified and revised as necessary during the process of extracting data from each included evidence source. Modifications will be described in the scoping review. Any disagreements that arise between the reviewers will be resolved through discussion, or with an additional reviewer. If appropriate, authors of papers will be contacted to request missing or additional data. The preliminary data extraction tool, which will be translated into Covidence can be found in Table 3.

## Stage 5: Collating, summarizing, and reporting results

The PRISMA diagram will be used to illustrate the review process and delineate stages where studies are eliminated with reasons specified. A narrative summary will accompany the charted results and will describe how the results relate to the reviews objective and questions.

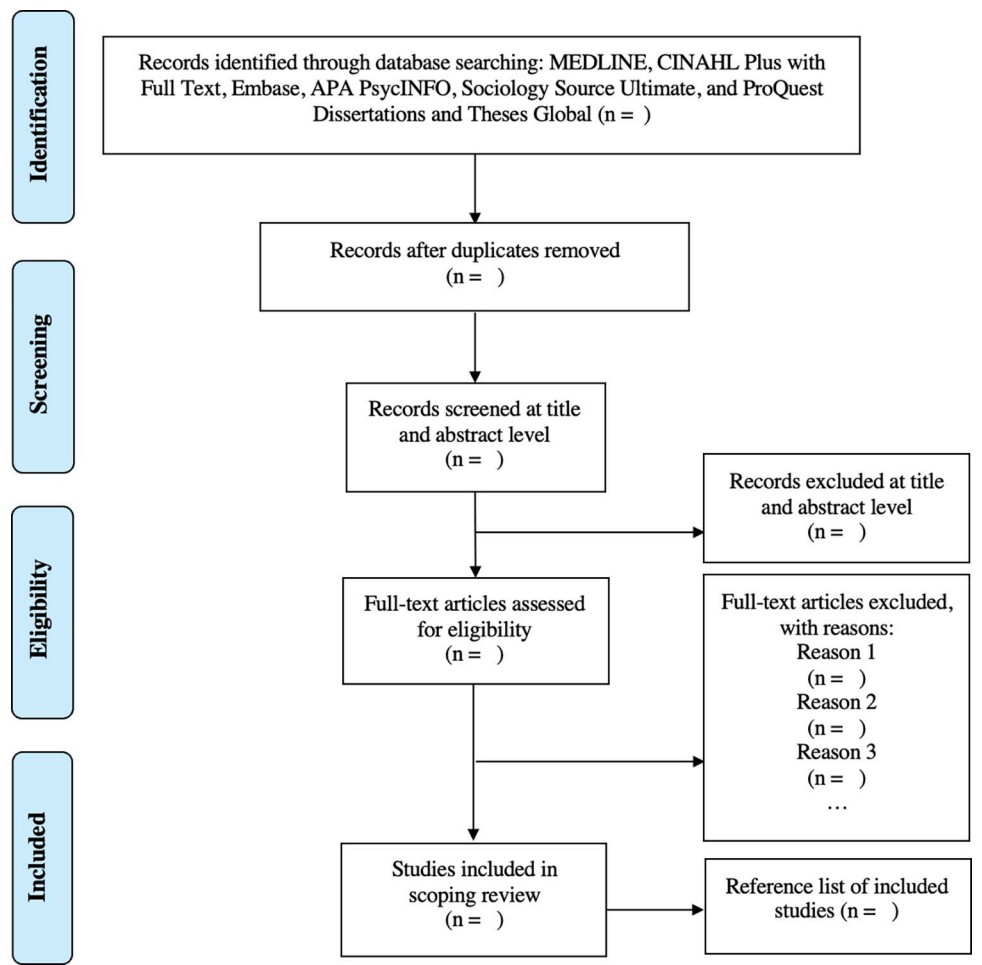

**Fig 1. PRISMA flow chart for the scoping review process.** Adapted from: Moher, Liberati, Tetzlaff, and Altman [25].

The results will be synthesized using the Caregiver Health Model (CGHM), which will provide a framework to categorize factors that impact caregivers' memory, including caregiver health promotion activities, caregiver attitudes and beliefs, caregiver task, and caregiver needs. Factors that do not fall into the Caregiver Health Model domains will be organized by emerging themes. Recurring patterns will be summarized and discussed. The findings of the proposed review will be disseminated via relevant scientific conferences and peer-reviewed publication.

## Stage 6: Conducting consultation

The selection of experts will be guided by their research background, with a focus on experienced researchers specializing in cognition (particularly memory), caregivers, and scoping review methodology. We plan to engage these experts throughout the process to enhance the overall quality of the review. The consultations with methodology and content experts will take the form of focus group discussions. During these sessions, we will present and discuss our findings to seek additional insights. This collaborative approach aims to gather input on various aspects, such as determining the optimal organization of different manuscript sections and identifying specific measurement tools for different memory domains (e.g., working memory and sensory memory).

**Table 3. Data extraction instrument.**

| |
| --- |
| **Study characteristics** |
| Author(s) |
| Year |
| Country |
| Study type/design |
| Study setting (hospital, long-term care facility, home, etc.) |
| Study purpose |
| Theoretical framework |
| Total number of participants |
| **Caregiver characteristics** |
| Total number of caregivers |
| Caregiver role/relationship to patient (s) |
| Caregiver age range |
| Caregiver race and ethnicity |
| Caregiving tasks/intensity/duration of caregiving activity |
| **Care-recipient characteristics** |
| Total number of care-recipients |
| Care-recipient diagnosis/severity |
| Care-recipient age range (pediatric vs. non-pediatric) |
| Care-recipient level of dependency/condition severity |
| **Caregiver memory** |
| Memory type |
| Memory measurement tool or instrument |
| Psychometrics of measurement tools/instruments (if applicable) |
| Memory measurement (result) |
| **Factors related to caregiver memory** |
| Factor name |
| Factor measurement tool or instrument |
| Psychometrics of measurement tools/instruments (if applicable) |
| Measurement time points (if applicable) |
| Analytical approach (if applicable) |
| Factor measurement (result) |
| **The main caregiver memory-related results of the study** |

## Discussion

The primary goal of the proposed scoping review is to identify the available evidence about memory in informal caregivers and understand the factors within the context of caregiving that are related to memory changes among caregivers. In addition to understanding the factors related to caregiver memory, obtaining a list of measurement tools of caregiver memory serves as a foundational resource for caregiver cognition studies. This initiative sets the stage for future research endeavors, such as a systematic review comparing the efficacy of various memory assessment tools in the caregiver population. It may also pave the way for studies focused on developing memory assessment tools tailored to the unique needs of caregivers. Moreover, this review explores potential evidence that may contribute to the development of interventions aimed at preventing or reducing cognitive problems in caregivers, potentially leading to improvements in caregiver functioning and care receiver health outcomes.

To ensure the reproducibility of the study, detailed plan for conducting the review is described in the study protocol. The potential limitation of the scoping review includes that

only evidence related to the memory domain of cognition will be assessed. Despite the limitations, to our knowledge, this is the first scoping review that intends to comprehensively map factors related to caregiver memory in the literature. The strengths of the scoping review include the inclusion of six major databases and a wide range of literature, including gray literature. Any changes to the study protocol will be reported in the scoping review, as well as the discussion of the limitations of the scoping review process. The findings will highlight research gaps that are relevant to caregiver memory. The results will be helpful for relevant stakeholders in developing guidelines for caregiver health promotion programs.

## Supporting information

**S1 Checklist. PRISMA-P checklist.** Adapted from: Shamseer et al. [26].
(DOC)

**S2 Checklist. PRISMA-ScR checklist.** Adapted from: Tricco et al. [22].
(DOCX)

## Acknowledgments

This review is to contribute towards the Nursing Ph.D. degree for DW.

## Author Contributions

**Conceptualization:** Dingyue Wang, Sharron Rushton, Kimberly Ramos, Cristina C. Hendrix.

**Investigation:** Dingyue Wang, Sharron Rushton, Kimberly Ramos, Cristina C. Hendrix.

**Methodology:** Dingyue Wang, Sharron Rushton, Leila Ledbetter, Margaret Graton, Cristina C. Hendrix.

**Project administration:** Dingyue Wang.

**Supervision:** Dingyue Wang, Sharron Rushton, Cristina C. Hendrix.

**Writing – original draft:** Dingyue Wang, Leila Ledbetter, Margaret Graton.

**Writing – review & editing:** Dingyue Wang, Sharron Rushton, Kimberly Ramos, Cristina C. Hendrix.

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
