## [Decision Letter · Decision Letter 0]

5 Oct 2023

PONE-D-23-20449Factors associated with memory of informal caregivers: A scoping review protocolPLOS ONE

Dear Dr. Wang,

Thank you for submitting your manuscript to PLOS ONE. After careful consideration, we feel that it has merit but does not fully meet PLOS ONE’s publication criteria as it currently stands. Therefore, we invite you to submit a revised version of the manuscript that addresses the points raised during the review process.

In particular, both Reviewer 1 and Reviewer 3 are concerned about the multiple inconsistencies between the methodology and the framework/guideline. Specifically, the data extraction approach is inconsistent with conceptual framework, which seems outdated. The inclusion/exclusion criteria need refining, and rationales need clearly explained. Please thoroughly address the reviewers' comments including those regarding typos and grammar. Please submit your revised manuscript by Nov 19 2023 11:59PM. If you will need more time than this to complete your revisions, please reply to this message or contact the journal office at plosone@plos.org. Please include the following items when submitting your revised manuscript:A rebuttal letter that responds to each point raised by the academic editor and reviewer(s). You should upload this letter as a separate file labeled 'Response to Reviewers'.A marked-up copy of your manuscript that highlights changes made to the original version. You should upload this as a separate file labeled 'Revised Manuscript with Track Changes'.An unmarked version of your revised paper without tracked changes. You should upload this as a separate file labeled 'Manuscript'.

We look forward to receiving your revised manuscript.

Kind regards,

Weifeng Han, PhD

Academic Editor

PLOS ONE

Reviewers' comments:

Reviewer's Responses to Questions

**Comments to the Author**

1. Does the manuscript provide a valid rationale for the proposed study, with clearly identified and justified research questions?

Reviewer #1: Yes

Reviewer #2: Yes

Reviewer #3: Yes

2. Is the protocol technically sound and planned in a manner that will lead to a meaningful outcome and allow testing the stated hypotheses?

Reviewer #1: Partly

Reviewer #2: Yes

Reviewer #3: Partly

3. Is the methodology feasible and described in sufficient detail to allow the work to be replicable?

Reviewer #1: Yes

Reviewer #2: Yes

Reviewer #3: No

4. Have the authors described where all data underlying the findings will be made available when the study is complete?

Reviewer #1: No

Reviewer #2: Yes

Reviewer #3: No

5. Is the manuscript presented in an intelligible fashion and written in standard English?

Reviewer #1: Yes

Reviewer #2: Yes

Reviewer #3: Yes

6. Review Comments to the Author

You may also provide optional suggestions and comments to authors that they might find helpful in planning their study.

Reviewer #1: The abstract is clearly written and provides an adequate summary of the submitted work.

The introduction is well-written and adequately introduces the challenges faced by caregivers, in particular cognitive decline, backed up by relevant and recent literature. However, geographical focus seems to be limited to the US, which is not appropriate as the scoping review will be a global review. Therefore, an overview of caregiving at the global level should be added to the introduction (not necessarily a comprehensive overview, but at least mention the findings and cite examples of studies from other parts of the world). Furthermore, at the end of the introduction a transitional paragraph that explains the rationale for the design and conduct of a scoping review is lacking. It seems that the paragraphs in lines 94-111 should logically follow at this point (before introducing the conceptual framework).

The CGHM serves as a suitable conceptual framework for the scoping review. A more detailed explanation of each of the independent constructs in this part of the paper is recommended though. The paragraphs in lines 94-111 would be better fit after line 84 (see comment above).

In lines 120 ff., what is the rationale for examining how caregiver memory is measured? This appears to not have been clearly justified in the preceding sections.

In lines 129 ff. and Table 1, the inclusion and exclusion criteria are mostly clearly explained. However, the focus on quantitative studies (which was mentioned elsewhere) is not reflected here. The search strategies are comprehensively documented. However, for the non-expert reader, accessibility would be improved if some of the search queries would be elaborated (e.g. in Table 2, what does “S1 OR S2 OR S3” mean). The search queries are presented in a quite technical format, which is appropriate but at the same time not enough to enable non-expert reader to easily understand this scoping review protocol.

In lines 170 ff. and Table 3, the data extraction approach is clearly documented. However, the Factor types (physical/psychosocial/demographic/…) do not seem to be consistent with the ones in the conceptual framework (caregiver health promotion activities, caregiver attitudes and beliefs, …). Why is that? More clarification on this might be necessary. Moreover, I strongly recommend to add a further row to Table 3 to extract the methods used to estimate the effects of factors on caregiver memory. Currently, only the measurement tools or instruments will be extracted, but not the quantitative analytical approach (bivariate vs multivariate analysis, correlational statistical methods used).

In lines 189 ff., the approach of collating, summarizing and reporting the results is well-described.

In lines 198 ff., it is not clear what the specific purpose of the consulting experienced research will be. What will be there role? How specifically will they improve this review?

The discussion in lines 202 ff. is mostly fine. An additional limitation is the neglect of qualitative studies.

Reviewer #2: Have the authors considered the other electronic databases besides the five databases mentioned in the manuscript? If no, please reconsider.

Reviewer #3: Dear authors

Thank you for the opportunity to review your protocol. The protocol considers the key components of a protocol and is well structured.

There are, however, some inconsistencies in the protocol and aspects of methodology that could lead to bias. The main inconsistencies exist between the objectives and the inclusion/exclusion criteria, the type of studies for inclusion in the methods section and those referred to in the discussion and between the abstract and the main body in terms of the primary approach to analysis. I feel some of the concepts need to be clarified and/justified, for example, what do you mean by chronic conditions, why adults are defined at age 21 for inclusion and why caregivers of both adults and children are included in the same review.

It is unclear why Arksey and O’Malley’s framework 2005 framework was selected to guide the conduct of the review when there is a 2020 guideline available and referred to within the paper. The landscape has changed a lot in how to conduct reviews in the last 15 years and it would be expected that more up-to-date guidelines would guide the conduct of review and/justification provided for the use of older guidelines.

Below are some specific examples of the limitations noted in the submitted protocol

1. On page 4 lines 91-93 it indicates that the emerging themes will be secondary to the categorisation of the findings using the caregiver health model. On page 2 the thematic analysis is presented as the primary approach to analysis.

2. The abstract states “This scoping review intends to comprehensively map factors related to caregiver memory reported in the literature within the chronic caregiving context. Specific aims include (1) identifying factors related to caregiver memory; (2) examining how caregiver memory has been measured; and (3) describing changes in caregiver memory during their caregiving period”.

However, Page 4: line 99-101 indicates that the review will also focus on relationships between variables. It is not clear to me which objective this relates to.

3. Page 4: lines 121 -123 on page 5: Why is it necessary to have both aims and objective and review questions in the scoping review? There appears to be unnecessary duplication.

4. Page 5: This scoping review protocol is registered to OSF , should this be the title and abstract as it is unusual to publish the same protocol in two places.

5. Page 6., lines 131-137: Mixing of tenses. States that the databases will be searched but later states that the search has been conducted.

6. 114-146: Earlier it was stated that data was to be reported quantitatively but some of the studies included are unlikely to provide quantitative data e.g. case studies,

7. Line 147: typo, on should be one?

8. Page 7, lines 155-156: It is unusual to amend the inclusion/exclusion criteria when the screening has commenced. It is usual that the inclusion/exclusion criteria would be well considered in advance as to amend during screening even at the pilot phase could result in bias.

9. Queries specific to Table 1:

Do patients include children as well as adults and if so how will the impact of that be considered in the analysis and write up of the findings and conclusions.

Why is adult defined as 21 for the inclusion criteria?

Part of the inclusion criteria is that (3) the study analysed if there was a relationship between caregiver memory and memory related factors. Which objective does this relate to?

In the exclusion criteria you write “(3) the study did not analyse if there was a relationship between caregiver memory and memory-related factors”. What if the paper reports data relating to the other objectives?

10. Queries specific to Table 2: Was there an information specialist involved in the design and conduct of the search? If so has this person’s contribution been acknowledged? The search does not seem to include synonyms for caregiver and therefore some relevant papers may not be included in the review.

11. Discussion: Page 12 In the paper it states “This review will provide evidence to develop interventions preventing or reducing cognitive problems in caregivers, which could ultimately improve caregiver functioning and care receiver health outcomes”. As the outcomes of the review are unknown it is not possible to be this definite as to the findings. A more tentative statement such as may provide evidence would be a more accurate representation of what is likely to emerge from the review findings.

Page 12, line 212: The inclusion of qualitative studies and opinion papers etc is inconsistent with that proposed in the methodology section and the type of data that will be extracted.

7. PLOS authors have the option to publish the peer review history of their article (what does this mean?). If published, this will include your full peer review and any attached files.

Reviewer #1: No

Reviewer #2: No

Reviewer #3: **Yes: **Margarita Corry

---

## [Author Response · Author response to Decision Letter 0]

19 Nov 2023

Responses to editor’s comments: 

Editor comments: Both Reviewer 1 and Reviewer 3 are concerned about the multiple inconsistencies between the methodology and the framework/guideline. Specifically, the data extraction approach is inconsistent with conceptual framework, which seems outdated. The inclusion/exclusion criteria need refining, and rationales need clearly explained. 

Address: Thank you for offering an overview and summary of the reviewer comments! Each component in the conceptual framework has been thoroughly defined. Each component in the conceptual framework have now been matched with the methodology. The data extraction approach has been revised to fit both the aims of the study as well as the components in conceptual framework. Incorporated both feedback from reviewer 1 and 3, our team has decided to include qualitative studies and adjust the inclusion and exclusion criteria accordingly. Due to this major revision, the PRISMA flow diagram and checklists are revised accordingly. In addition, the age range and reasons of including both caregivers for adults and children are thoroughly explained. Details are included in the Response to Reviewers section. 

Editor comments: Please thoroughly address the reviewers' comments including those regarding typos and grammar.

Address: Each typo and grammar mistakes are carefully address in the manuscript. 

 

Responses to reviewer #1: 

Reviewer comments: The geographical focus seems to be limited to the US, which is not appropriate as the scoping review will be a global review. Therefore, an overview of caregiving at the global level should be added to the introduction (not necessarily a comprehensive overview, but at least mention the findings and cite examples of studies from other parts of the world)

Address: We agree that incorporating a global perspective is crucial. Initially, we considered including a specific global statistic on the caregiver population, stating, "In 2018, there were 647 million full-time caregivers worldwide (Barnes & Ramanarayanan, 2022)." However, recognizing the limitation of the data being five years old and the challenges in obtaining more recent global caregiver population data, potentially influenced by the ongoing COVID-19 pandemic, we opted for a more general statement. In the introduction, we now included, “Caregiving is an important public health issue, and the escalating need for care is a global concern due to various factors, such as a rising population living into advanced ages and a consequent surge in individuals with chronic conditions (Haley & Elayoubi, 2023).” In addition, our introduction encompasses articles on caregiver cognition from England (García-Castro et al., 2022), Canada (Mallya et al., 2018), and South Korea (Yang et al., 2021) (Changes made in Line 59-61 in the revised manuscript). 

Reference:

García-Castro FJ, Bendayan R, Dobson RJ, Blanca MJ. Cognition in informal caregivers: Evidence from an English population study. Aging Ment Health. 2022;26(3):507-18. 

Haley, W. E., & Elayoubi, J. (2023). Family caregiving as a global and lifespan public health issue. The Lancet. Public health, S2468-2667(23)00227-X. Advance online publication. https://doi.org/10.1016/S2468-2667(23)00227-X

Mallya S, Fiocco AJ. Impact of informal caregiving on cognitive function and well-being in Canada. International Psychogeriatrics. 2018;30(7):1049-55. 

Yang, H. W., Bae, J. B., Oh, D. J., Moon, D. G., Lim, E., Shin, J., Kim, B. J., Lee, D. W., Kim, J. L., Jhoo, J. H., Park, J. H., Lee, J. J., Kwak, K. P., Lee, S. B., Moon, S. W., Ryu, S. H., Kim, S. G., Han, J. W., & Kim, K. W. (2021). Exploration of cognitive outcomes and risk factors for cognitive decline shared by couples. JAMA Network Open, 4(12), e2139765. https://doi.org/10.1001/jamanetworkopen.2021.39765

Barnes, S. B. & Ramanarayanan, D. (2022, April). Global health & gender policy brief: The global care economy. Wilson center. https://www.wilsoncenter.org/publication/global-health-gender-policy-brief-global-care-economy#:~:text=Globally%2C%20647%20million%20full%2Dtime,caregivers%20prior%20to%20the%20pandemic.

Reviewer comments: At the end of the introduction a transitional paragraph that explains the rationale for the design and conduct of a scoping review is lacking. It seems that the paragraphs in lines 94-111 should logically follow at this point (before introducing the conceptual framework). The CGHM serves as a suitable conceptual framework for the scoping review. A more detailed explanation of each of the independent constructs in this part of the paper is recommended though. The paragraphs in lines 94-111 would be better fit after line 84 (see comment above).

Address: It is very helpful feedback! The paragraphs in lines 94-111 (in original manuscript) is now moved after line 84 (in original manuscript) for a better logical flow (changes made in Line 89-115 in the revised manuscript). A detailed explanation of each of the independent constructs in CGHM is added (changes made in Line 103-112 in the revised manuscript). 

In the present study, “caregiver health promotion activity” is defined as the strategies caregivers take to maintain or improve their memory (e.g., physical exercise, diet) and interventions or programs designed for caregivers that are related to caregiver memory (e.g., caregiver training program). “Caregiver attitude and belief” is defined as caregivers' subjective evaluations and cognitive stances towards their caregiving responsibilities, the care recipients (patients), and the broader healthcare environment (e.g., cognitive stress appraisal). “Caregiver tasks” is defined as the activities (e.g., assisting with Activities of Daily Living) or workload (e.g., hours of caregiving per day) that caregivers have that are related to caregiver memory. Finally, "caregiver needs" is defined as the specific requirements, challenges, and support mechanisms required by caregivers that is related to caregiver memory (e.g., emotional support needs, respite care needs).

Reviewer comments: In lines 120 ff., what is the rationale for examining how caregiver memory is measured? This appears to not have been clearly justified in the preceding sections.

Address: This is a great point that we need to provide the rationale for examining how caregiver memory is measured! 

Caregiver memory is an underexplored area. Scoping review provides a broad overview and mapping of existing literature on a particular topic, and often serves as a great starting point in an area of research. Obtaining a list of measurement tools of caregiver memory serves as a foundational resource for future caregiver cognition studies. This initiative sets the stage for potential future research endeavors, such as a systematic review comparing the efficacy of various memory assessment tools in the caregiver population. It may also pave the way for studies focused on developing memory assessment tools tailored to the unique needs of caregivers, including potential psychometric investigations. Moreover, understanding how caregiver memory is measured is a crucial step in initiating caregiver memory intervention studies. 

For the flow of the paragraphs, the intention of including memory measurement has been briefly introduced in the introduction (Line 94 in the revised manuscript), and the rationale is described in detail in the discussion section (Line 218-223 in the revised manuscript). 

Reviewer comments: In lines 129 ff. and Table 1, the inclusion and exclusion criteria are mostly clearly explained. However, the focus on quantitative studies (which was mentioned elsewhere) is not reflected here.

Address: Thanks for your insightful comments! Identifying this methodology inconsistency is a valuable observation for us. In response to this concern (plus another reviewer's mentioned the limitation of excluding qualitative studies), our team has decided to include qualitative studies and adjust the inclusion and exclusion criteria accordingly. Specifically, the inclusion and exclusion criteria are change into:

“Studies will be included if the study (1) provides information pertaining to factors related to memory of informal caregivers; (2) includes caregiver memory measurement tool (s); and (3) describes changes in caregiver memory during their caregiving period. Studies will be excluded if (1) caregiver memory is not measured and (2) the study solely focuses on other cognitive variables, outside of memory.”

This adjustment aims to better align with the scoping review's purpose, seeking to encompass a broad range of literature on the topic. By minimizing restrictions on inclusion and exclusion criteria, we aim to enhance the methodology's ability to capture a diverse array of findings. This approach is intended to reduce the likelihood of overlooking unexpected results and, consequently, mitigate potential biases. Your input has been instrumental in refining our methodology (Changes made in Line 174 (Table 1) in the revised manuscript).

Reviewer comments: The search strategies are comprehensively documented. However, for the non-expert reader, accessibility would be improved if some of the search queries would be elaborated (e.g., in Table 2, what does “S1 OR S2 OR S3” mean). The search queries are presented in a quite technical format, which is appropriate but at the same time not enough to enable non-expert reader to easily understand this scoping review protocol.

Address: Thanks for the feedback! Yes, it is important to consider non-expert readers. We have added a footnote under Table 3 (Changes made in Line 175 in the revised manuscript):

“S1 OR S2 means that the search results will include studies that match either S1 or S2 or both. It broadens the search to capture a larger set of relevant studies. S1 AND S2 means that the search results will include studies that satisfy both S1 and S2. This narrows down the search to find studies that meet both criteria.” 

However, it is difficult to change the search queries into a less technical format, which serve purpose for it to be replicable for researchers. As our team reviewed several sample papers published by PLOS ONE (Amri et al., 2022; Widyaningsih et al., 2022), a technical format is not discouraged. 

Reference:

Amri, M., Ali, S., Jessiman-Perreault, G., Barrett, K., & Bump, J. B. (2022). Evaluating healthy cities: A scoping review protocol. Plos one, 17(10), e0276179.

Widyaningsih, V., Febrinasari, R. P., Sari, V., Augustania, C., Verlita, B., Wahyuni, C., ... & Probandari, A. (2022). Potential and challenges for an integrated management of tuberculosis, diabetes mellitus, and hypertension: A scoping review protocol. Plos one, 17(7), e0271323.

Reviewer comments: In lines 170 ff. and Table 3, the data extraction approach is clearly documented. However, the Factor types (physical/psychosocial/demographic/…) do not seem to be consistent with the ones in the conceptual framework (caregiver health promotion activities, caregiver attitudes and beliefs, …). Why is that? More clarification on this might be necessary. Moreover, I strongly recommend adding a further row to Table 3 to extract the methods used to estimate the effects of factors on caregiver memory. Currently, only the measurement tools or instruments will be extracted, but not the quantitative analytical approach (bivariate vs multivariate analysis, correlational statistical methods used).

Address: Thanks for pointing out this issue, it is indeed very confusing to readers. We will stick to the CGHM conceptual framework. Factors related to caregiver memory are grouped under caregiver health promotion activities, caregiver attitudes and beliefs, caregiver tasks, and caregiver needs. Because including the factor type is confusing, we removed the “physical/psychosocial/demographic/…” from Table 3. Also, great suggestion about adding the column for analytical approach! The additional column is added in Table 3 (Both changes made in Line 195 in the revised manuscript).

Reviewer comments: In lines 198 ff., it is not clear what the specific purpose of the consulting experienced research will be. What will be their role? How specifically will they improve this review?

Address: Your feedback is greatly appreciated! Conducting consultation is an optional step suggested by the widely used Arksey & O'Malley (2005) scoping review guideline. Consultation can serve various purposes, including assistance in determining the optimal organization of different manuscript sections. Content experts may provide valuable insights into which specific measurement tools measure particular domains of memory (e.g., working memory vs. sensory memory). It's important to highlight that while consultation is traditionally positioned as the final step in a review, it can be integrated throughout the review stages (Mak & Thomas, 2022). Throughout the entire review process, content experts as well as methodology experts’ input will be considered. They may not satisfy the criteria of a coauthor in the final scoping review manuscript, but their contribution will be stated in acknowledgement. We recognized that the information we included in the “Conducting Consultation” section lacks the depth and clarity.

To address this, we will enhance the section to articulate our approach more clearly. We revised this section into “The selection of experts will be guided by their research background, with a focus on experienced researchers specializing in cognition (particularly memory), caregivers, and scoping review methodology. We plan to engage these experts throughout the process to enhance the overall quality of the review. The consultations with methodology and content experts will take the form of focus group discussions. During these sessions, we will present and discuss our findings to seek additional insights. This collaborative approach aims to gather input on various aspects, such as determining the optimal organization of different manuscript sections and identifying specific measurement tools for different memory domains (e.g., working memory vs. sensory memory).” (Changes made in Line 207-214 in the revised manuscript).

Reference:

Arksey, H., & O'Malley, L. (2005). Scoping studies: towards a methodological framework. International journal of social research methodology, 8(1), 19-32.

Mak, S., & Thomas, A. (2022). Steps for Conducting a Scoping Review. Journal of graduate medical education, 14(5), 565–567. https://doi.org/10.4300/JGME-D-22-00621.1

Reviewer comments: The discussion in lines 202 ff. is mostly fine. An additional limitation is the neglect of qualitative studies.

Address: Thanks for the comments, it is indeed a limitation! As another reviewer also brought up the same issue, our team made the decision to also incorporate qualitative studies and adjust the inclusion and exclusion criteria accordingly (Changes made in Line 174 (Table 1) in the revised manuscript). We believe this modification aligns better with the scoping review's purpose, aiming to encompass a broad range of literature on the topic. By minimizing restrictions on inclusion and exclusion criteria, we aim to enhance the methodology's capacity to capture a diverse array of findings. This approach is intended to reduce the likelihood of overlooking unexpected results and, consequently, mitigate potential biases.

 

Responses to reviewer #2: 

Reviewer comments: Have the authors considered the other electronic databases besides the five databases mentioned in the manuscript? If no, please reconsider.

Address: Thanks for the comments! We plan to conduct search in six electronic databases, including MEDLINE (via Ovid), CINAHL Plus with Full Text (via EBSCOhost), Embase (via Elsevier), APA PsycINFO (via EBSCOhost), Sociology Source Ultimate (via EBSCOhost), and ProQuest Dissertations and Theses Global. These choices could cover a wide range of literature, including grey literature, as we can find dissertations using ProQuest, and we can find conference proceedings using Embase (Changes made in Line 141 in the revised manuscript).

 

Responses to reviewer #3: 

Reviewer comments: The main inconsistencies exist between the objectives and the inclusion/exclusion criteria, the type of studies for inclusion in the methods section and those referred to in the discussion and between the abstract and the main body in terms of the primary approach to analysis.

Address: Thanks for providing a summary of the main revision point! Each specific point was addressed below.

Reviewer comments: I feel some of the concepts need to be clarified and/justified, for example, what do you mean by chronic conditions, why adults are defined at age 21 for inclusion and why caregivers of both adults and children are included in the same review.

Address: 

1. We fully acknowledge the importance of clearly defining the terms in scoping review protocol! Defining chronic conditions has posed a challenge due to significant variations within professional communities such as medical, public health, academia, and policy. For example, the CDC classify the following as chronic diseases: heart disease, stroke, cancer, type 2 diabetes, obesity, and arthritis. The Centers for Medicare and Medicaid Services have a more extensive list of 19 chronic conditions that includes Alzheimer’s disease, depression, HIV, etc. (Bernell & Howard, 2016). Moreover, there is a growing inclination to encompass chronic conditions that extend beyond traditional disease indicators, including long-standing functional disabilities such as developmental disorders (Bernell & Howard, 2016). The dynamic nature of diseases, transitioning from acute or fatal to chronic, further complicates the distinction between chronic and non-chronic conditions. Given that our study primarily focuses on caregiver memory and memory change, we plan to include caregivers who have provided care for a prolonged duration (defined as more than 6 months in this study). We will not exclude studies based on patient diagnoses, as we recognize the potential impact of patient diagnosis on caregiver memory. In this study, we adopt a broad definition of chronic conditions as any disease with a prolonged duration.

2. The revision and rationale of age cutoff (we changed from 21 to 18) is explained in details below. 

3. To explain the rationale for including caregivers of both adults and children in this review, it's crucial to recognize that many grandparents assume caregiving responsibilities for their grandchildren experiencing chronic conditions. Given that grandparents are typically at an age where memory concerns may arise, including caregivers of both adults and children adds valuable diversity to our study. If significant differences emerge between the experiences of caregivers for adults versus those caring for children, our plan is to allocate a paragraph in both the Results and Discussion sections to elucidate and discuss these distinctions.

Reference:

Bernell, S., & Howard, S. W. (2016). Use Your Words Carefully: What Is a Chronic Disease?. Frontiers in public health, 4, 159. https://doi.org/10.3389/fpubh.2016.00159

Reviewer comments: It is unclear why Arksey and O’Malley’s framework 2005 framework was selected to guide the conduct of the review when there is a 2020 guideline available and referred to within the paper. The landscape has changed a lot in how to conduct reviews in the last 15 years and it would be expected that more up-to-date guidelines would guide the conduct of review and/justification provided for the use of older guidelines.

Address: It is a very insightful question that you brought up! The authors had several rounds of discussion related to the framework choice for the scoping review, and we still plan to use Arksey and O’Malley’s framework 2005 for the following reasons: 

1. The new JBI guideline (Peters et al., 2021) is not technically a step-by-step guidance for scoping review. Unlike Arksey and O’Malley’s framework, which specified the required steps (as well as optional steps) for scoping review, Peters et al., (2021) focused more on stating which components are necessary, which are (1) Title and review questions, (2) Inclusion criteria (Participants, Concept, Context), (3) Types of evidence sources, (4) Search strategy, (5) Evidence screening and selection, (6) Data extraction, (7) Data analysis, and (8) Presentation of results. Peters et al., (2021) did not specify the standard/expected way of how to organize these components. 

2. Arksey and O’Malley’s framework is a widely recognized and influential approach in the field of research methodology. It has been adopted and adapted by researchers in various disciplines due to its utility in addressing research questions where a broad overview of the existing literature is needed. As the field of research methodology evolves, scholars continue to build upon and refine scoping review approaches, but Arksey and O'Malley's framework still remains a seminal and enduring contribution to the methodology of literature reviews. 

3. Although the landscape has changed in the last 15 years, Arksey and O’Malley’s framework is not outdated. A more recent scoping review guideline suggested the same steps as Arksey and O’Malley’s framework: (1) identifying the research question, (2) identifying relevant studies, (3) study selection, (4) charting the data, (5) collating, summarizing, reporting results, and (6) conducting consultation (Mak & Thomas, 2022). The only difference is the wording of step (3) and (6). In the 2022 framework, step (3) is called “Selecting Studies to Be Included in the Review” and step (6) is called “Consulting Stakeholders.”

4. Although we chose Arksey and O’Malley’s framework, we still included the unique part (something not in Arksey and O’Malley’s framework) of Peters et al., (2021)’s guideline, which is the Participants, Concept, Context (PCC) framework to guide the inclusion and exclusion criteria. 

Reference:

Mak, S., & Thomas, A. (2022). Steps for Conducting a Scoping Review. Journal of graduate medical education, 14(5), 565–567. https://doi.org/10.4300/JGME-D-22-00621.1

Peters, M. D., Marnie, C., Tricco, A. C., Pollock, D., Munn, Z., Alexander, L., ... & Khalil, H. (2021). Updated methodological guidance for the conduct of scoping reviews. JBI evidence implementation, 19(1), 3-10.

Reviewer comments: On page 4 lines 91-93 it indicates that the emerging themes will be secondary to the categorisation of the findings using the caregiver health model. On page 2 the thematic analysis is presented as the primary approach to analysis.

Address: This comment is extremely helpful! Compared to page 2 (abstract), Page 4 (lines 91-93 in original manuscript) is a more accurate description for the method. As you nicely summarized: emerging themes will be secondary to the categorization of the findings using the Caregiver Health Model. The last part of abstract is changed to “The caregiver health model will provide a framework to categorize factors that impact caregivers’ memory including caregiver health promotion activities, caregiver attitudes and beliefs, caregiver task, and caregiver needs as proposed by Caregiver Health Model. Factors that do not fall into the Caregiver Health Model domains will be organized by emerging themes.” (Changes made in Line 46-49 in the revised manuscript). 

Reviewer comments: The abstract states “This scoping review intends to comprehensively map factors related to caregiver memory reported in the literature within the chronic caregiving context. Specific aims include (1) identifying factors related to caregiver memory; (2) examining how caregiver memory has been measured; and (3) describing changes in caregiver memory during their caregiving period”. However, Page 4: line 99-101 indicates that the review will also focus on relationships between variables. It is not clear to me which objective this relates to.

Address: This observation is very helpful! As we read the aims again, we also noticed this inconsistency. The major aims will still be (1) identifying factors related to caregiver memory; (2) examining how caregiver memory has been measured; and (3) describing changes in caregiver memory during their caregiving period. We initially assumed that for those articles that provide us information about factors related to caregiver memory, will also reveal some relationships between those variables. But indeed, exploring the relationship between caregiving and caregiver memory is not the main focus, and this information did make line 99-101 (in original manuscript) confusing to readers. To make the objectives clear and concise, we removed the relationship piece from the protocol (Changes made in Line 174 (Table 1) in the revised manuscript).

Reviewer comments: Page 4: lines 121 -123 on page 5: Why is it necessary to have both aims and objective and review questions in the scoping review? There appears to be unnecessary duplication.

Address: Thank so much for bring up this great point here! I removed the specific research questions and only kept the aims, as I checked the Submission Guidelines, aims are required, research questions are not. (Changes made in Line 137 in the revised manuscript).

https://journals.plos.org/plosone/s/submission-guidelines#loc-study-protocols

Reviewer comments: Page 5: This scoping review protocol is registered to OSF, should this be the title and abstract as it is unusual to publish the same protocol in two places.

Address: I appreciate the feedback and understand the concern here! However, PLOS ONE encourages authors to register with OSF and provide the registration number in the Materials and Methods section. Protocol registration is a best practice recommended by major guidelines such as the JBI Manual. Our team did not post the entire protocol in OSF, as you mentioned, only the title and abstract are currently available in OSF. Protocol registration reduces bias by establishing the criteria a priori, promotes transparent research methodology, and reduces duplication of efforts. There is a distinction between publication and registration, and OSF is the most common place to deposit protocols for scoping reviews. 

https://guides.atsu.edu/scopingreviews/registration

Reviewer comments: Page 6., lines 131-137: Mixing of tenses. States that the databases will be searched but later states that the search has been conducted.

Address: Thanks for bringing up this inconsistency! The sentence “Literature search has been completed” has been removed from the protocol (Changes made in Line 148 in the revised manuscript).

Additional explanation: Although we have conducted a preliminary search (a total of 7,578 citations were identified), we intend to update the search as the project progresses. It is important to highlight that the literature search is still in progress and the screening process has not yet commenced.

Reviewer comments: 114-146: Earlier it was stated that data was to be reported quantitatively but some of the studies included are unlikely to provide quantitative data e.g. case studies.

Address: As another reviewer also brought up a concern related to the inclusion of qualitave versus quantitative studies, our team made the decision to also incorporate qualitative studies and adjust the inclusion and exclusion criteria accordingly (Changes made in Line 174 (Table 1) in the revised manuscript). We believe this modification aligns better with the scoping review's purpose, aiming to encompass a broad range of literature on the topic. By minimizing restrictions on inclusion and exclusion criteria, we aim to enhance the methodology's capacity to capture a diverse array of findings. This approach is intended to reduce the likelihood of overlooking unexpected results and, consequently, mitigate potential biases.

Reviewer comments: Line 147: typo, on should be one?

Address: Thanks for catching the typo! This sentence is deleted based on another feedback. 

Reviewer comments: Page 7, lines 155-156: It is unusual to amend the inclusion/exclusion criteria when the screening has commenced. It is usual that the inclusion/exclusion criteria would be well considered in advance as to amend during screening even at the pilot phase could result in bias.

Address: Thanks for the comment! I agree that it is unusual to amend the inclusion/exclusion criteria once the screening has commenced. However, we need to pilot test at various stages in the scoping review process. Piloting of search strategies, piloting of inclusion/exclusion criteria (within the screening stage) and piloting of data extraction tables are all regularly employed by scoping reviewers (Long, 2014). Insights from pilot review may be used to modify the inclusion/exclusion criteria (Long, 2014). However, amending inclusion/exclusion criteria is not the only goal of pilot testing, we also intend to see if the eligibility criteria are expressed clearly enough, if the screeners on the review team interpret the criteria consistently, and if there are articles which should be included in the review but had not been anticipated as being relevant? 

The statement "Screeners will pilot 25 titles and abstracts and refine inclusion and exclusion criteria as needed. Following a pilot test, titles and abstracts will be screened by two independent reviewers for assessment against the inclusion criteria." has been changed to " Screeners will pilot 25 titles and abstracts to assess the clarity of eligibility criteria, the consistency of criteria interpretation by screeners on the review team, and the need for refinement in inclusion and exclusion criteria." (Line 164-166 in the revised manuscript).

Reference:

Long L. (2014). Routine piloting in systematic reviews--a modified approach?. Systematic reviews, 3, 77. https://doi.org/10.1186/2046-4053-3-77

Reviewer comments: Queries specific to Table 1: Do patients include children as well as adults and if so how will the impact of that be considered in the analysis and write up of the findings and conclusions. Why is adult defined as 21 for the inclusion criteria?

Address: Thank you for your inquiry regarding the inclusion of both children and adults in our study. Indeed, our patient population is not limited by age, encompassing both pediatric and adult cases. We assure that the impact of this diverse patient demographic on the analysis and write-up will be meticulously considered. In the event that we observe notable differences in caregiver memory concerning those caring for children versus adults, we plan to dedicate a paragraph in the Results section and another in the Discussion section to address these distinctions. It's important to note that, as of now, the review has not been conducted, and we do not hold any a priori assumptions regarding the specific details of the analysis and write-up. We remain committed to thoroughly exploring and reporting any pertinent findings related to the diverse age range of our patient population.

For the age inclusion criteria, we did have several rounds of conversation among authors to determine the best age cutoff. This study aims to focus on adult caregivers. Although teenagers or children also sometimes contribute to long term caregiving for chronic illnesses, they have distinctive experience related to caregiving and unique cognitive processes compared to adult caregivers and will not be the focus of the present study (Pakenham et al., 2006; Shifren & Kachorek, 2003). However, the age cutoff of adult caregivers remains inconsistent across literature. There are studies suggested that 18 is the cutoff of adult caregivers (Pope et al., 2022; Chevrier et al., 2022). Another two studies used 21 years old as cutoff for their inclusion criteria of young caregivers (Shifren & Kachorek, 2003; Shifren, 2001). Overall, given the rationale above, there is no overarching reason of choosing a specific age cutoff. 

Your insightful comment has prompted a reconsideration of the age criteria. After careful deliberation, we have decided to go for the lower age cutoff of 18 years old. This decision is aimed at ensuring a broader inclusion of articles, as unexpected findings related to the memory of younger caregivers may emerge during the review process. We appreciate your valuable input, which has contributed to refining our approach to age inclusion in the study (Changes made in Line 41 and 174 (Table 1) in the revised manuscript).

Reference:

Chevrier, B., Lamore, K., Untas, A., & Dorard, G. (2022). Young adult carers' identification, characteristics, and support: A systematic review. Frontiers in psychology, 13, 990257. https://doi.org/10.3389/fpsyg.2022.990257

Pakenham, K. I., Bursnall, S., Chiu, J., Cannon, T., & Okochi, M. (2006). The psychosocial impact of caregiving on young people who have a parent with an illness or disability: Comparisons between young caregivers and noncaregivers. Rehabilitation Psychology, 51(2), 113.

Pope, N. D., Baldwin, P. K., Gibson, A., & Smith, K. (2022). Becoming a Caregiver: Experiences of Young Adults Moving into Family Caregiving Roles. Journal of adult development, 29(2), 147–158. https://doi.org/10.1007/s10804-021-09391-3

Shifren, K. (2001). Early caregiving and adult depression: Good news for young caregivers. The Gerontologist, 41(2), 188-190.

Shifren, K., & Kachorek, L. (2003). Does early caregiving matter? The effects on young caregivers' adult mental health. International Journal of Behavioral Development, 27(4), 338-346. https://doi.org/10.1080/01650250244000371

Reviewer comments: Queries specific to Table 1: Part of the inclusion criteria is that (3) the study analysed if there was a relationship between caregiver memory and memory related factors. Which objective does this relate to?

Address: Thanks for this thoughtful comment! As we read the aims again, we also realized this inconsistency. Our aims are (1) identifying factors related to caregiver memory; (2) examining how caregiver memory has been measured; and (3) describing changes in caregiver memory during their caregiving period. We initially assumed that for those articles that provide information about factors related to caregiver memory, will also reveal some relationships between those variables. But indeed, exploring the relationship between caregiving and caregiver memory is not the main focus and is confusing to readers. In order to fit the methodology with the aims, we removed this inclusion criterion from the manuscript (Changes made in Line 174 (Table 1) in the revised manuscript).

Reviewer comments: Queries specific to Table 1: In the exclusion criteria you write “(3) the study did not analyse if there was a relationship between caregiver memory and memory-related factors”. What if the paper reports data relating to the other objectives?

Address: Same as above, thanks for pointing out this inconsistency! It's true that a paper may still report data related to other objectives. We removed this exclusion criterion (Changes made in Line 174 (Table 1) in the revised manuscript).

Reviewer comments: Queries specific to Table 2: Was there an information specialist involved in the design and conduct of the search? If so has this person’s contribution been acknowledged? The search does not seem to include synonyms for caregiver and therefore some relevant papers may not be included in the review.

Address: Two medical librarians specialized in systematic/scoping reviews and meta-analysis were involved in the design and conduct of the search. The reason that they are not in the acknowledgement section is because they wrote a part of the method section and were listed as coauthors. It is a long and complicated Table, and it is easy to miss the synonyms. We did include synonyms in row 3, 4, 11, 12, 17, 19, etc. The synonyms of caregiver (s) include carer (s), caretaker (s), and care partner (s) (although in different format for different databases).

Reviewer comments: Discussion: Page 12 In the paper it states “This review will provide evidence to develop interventions preventing or reducing cognitive problems in caregivers, which could ultimately improve caregiver functioning and care receiver health outcomes”. As the outcomes of the review are unknown it is not possible to be this definite as to the findings. A more tentative statement such as may provide evidence would be a more accurate representation of what is likely to emerge from the review findings.

Address: Thank you for the feedback! Upon revisiting the sentence, we recognized the need for a softer tone. The definite wording has been adjusted to "This review explores potential evidence that may contribute to the development of interventions aimed at preventing or reducing cognitive problems in caregivers, potentially leading to improvements in caregiver functioning and care receiver health outcomes." Revised in the manuscript (Changes made in Line 223-226 in the revised manuscript).

Reviewer comments: Page 12, line 212: The inclusion of qualitative studies and opinion papers etc is inconsistent with that proposed in the methodology section and the type of data that will be extracted.

Address: This feedback is very helpful! Since our team decided to also include qualitative studies to better fit the methodology with the aims, the inclusion and exclusion criteria are revised, and this sentence has been removed in the revised manuscript. (Changes made in Line 174 (Table 1) in the revised manuscript).

 

Dear editors and reviewers,

Overall, the feedback has been proved exceptionally constructive, greatly contributing to the refinement of our original submission. We are genuinely appreciative of the effort invested in providing such insightful comments. Once again, we extend our sincere gratitude for your time and thoughtful consideration of our work.

Best regards

Dingyue Wang

---

## [Editor Report · Decision Letter 1]

22 Nov 2023

Factors associated with memory of informal caregivers: A scoping review protocol

PONE-D-23-20449R1

Dear Dr. Wang,

We’re pleased to inform you that your manuscript has been judged scientifically suitable for publication and will be formally accepted for publication once it meets all outstanding technical requirements.

Kind regards,

Weifeng Han, PhD

Academic Editor

PLOS ONE

Additional Editor Comments (optional):

Thanks very much for carefully considering the reviewers' feedback and made substantive revisions to improve the manuscript. I am happy with the content. Please proofread the revised manuscript, e.g., there are still some inconsistencies in verb tenses between past and future tense when describing planned methods/analyses vs what has already been completed.
---

## [Editor Report · Acceptance letter]

18 Jan 2024

PONE-D-23-20449R1 

PLOS ONE

Dear Dr. Wang, 

I'm pleased to inform you that your manuscript has been deemed suitable for publication in PLOS ONE. Congratulations! Your manuscript is now being handed over to our production team.

Kind regards, 

on behalf of

Dr. Weifeng Han 

Academic Editor

PLOS ONE